# Influenza Virus Inactivated by Heavy Ion Beam Irradiation Stimulates Antigen-Specific Immune Responses

**DOI:** 10.3390/pharmaceutics16040465

**Published:** 2024-03-27

**Authors:** Kai Schulze, Ulrich Weber, Christoph Schuy, Marco Durante, Carlos Alberto Guzmán

**Affiliations:** 1Department of Vaccinology and Applied Microbiology, Helmholtz Zentrum für Infektionsforschung (HZI), 38124 Braunschweig, Germany; carlosalberto.guzman@helmholtz-hzi.de; 2Biophysics Department, GSI Helmholtzzentrum für Schwerionenforschung, 64291 Darmstadt, Germany; u.weber@gsi.de (U.W.); c.schuy@gsi.de (C.S.); m.durante@gsi.de (M.D.); 3Fachbereich Mathematik, Naturwissenschaften und Informatik, Technische Hochschule Mittelhessen, 35390 Gießen, Germany; 4Institute of Condensed Matter Physics, Technische Universität Darmstadt, 64289 Darmstadt, Germany; 5Department of Physics “Ettore Pancini”, University Federico II, 80138 Naples, Italy

**Keywords:** influenza, vaccination, gamma rays, heavy ions

## Abstract

The COVID-19 pandemic has made clear the need for effective and rapid vaccine development methods. Conventional inactivated virus vaccines, together with new technologies like vector and mRNA vaccines, were the first to be rolled out. However, the traditional methods used for virus inactivation can affect surface-exposed antigen, thereby reducing vaccine efficacy. Gamma rays have been used in the past to inactivate viruses. We recently proposed that high-energy heavy ions may be more suitable as an inactivation method because they increase the damage ratio between the viral nucleic acid and surface proteins. Here, we demonstrate that irradiation of the influenza virus using heavy ion beams constitutes a suitable method to develop effective vaccines, since immunization of mice by the intranasal route with the inactivated virus resulted in the stimulation of strong antigen-specific humoral and cellular immune responses.

## 1. Introduction

As recently demonstrated during the COVID-19 pandemic, vaccines represent a very efficient tool for preventing severe outcomes of rapidly spreading respiratory pathogens, even if they are not able to block horizontal transmission to susceptible hosts. In this regard, significant progress has been made in vaccine development during the last decades. While most of the vaccines approved for human use represent live attenuated (e.g., measles, mumps, rubella and influenza) or inactivated (e.g., whooping cough, polio, and influenza) vaccines, newer approaches are based on selected subcellular components of the infectious agents (e.g., diphtheria, tetanus, *Bordetella pertussis*, papilloma virus, or hepatitis B virus) or the genetic information encoding for these antigens (e.g., vector and mRNA vaccines, like those for SARS-CoV-2, respiratory syncytial virus). While single antigens evoke immune responses focused directly on the molecule that is relevant for protection, the immune responses stimulated might not provide sustained protection against disease, requiring boosters, in particular for pathogens with genetic plasticity. On the other hand, vaccines encompassing completely intact (attenuated or inactivated) pathogens usually are much more reactogenic, and often promote broader and more prolonged protection [1]. Using only selected microbial components for subunit or RNA vaccines usually results in an increased safety profile compared to live-attenuated or inactivated vaccine formulations. However, this improvement in general comes at the expense of the immunogenicity of the vaccine, making booster immunizations or the inclusion of adjuvants essential. In the case of RNA vaccines, complex liponanoparticles are needed to protect the RNA cargo and to guarantee proper delivery. Furthermore, more stringent cold-chains are required and the costs are higher than for whole-cell vaccines. This is not a trivial issue since, as an example, despite the availability of effective subunit vaccines, attenuated vaccines and inactivated vaccines produced in cell factories, inactivated vaccines produced in egg-based factories still represent the most efficient approach to obtain high yields at an acceptable cost for addressing worldwide needs during the influenza seasons.

In this regard, inactivation of vaccine antigens usually is performed using chemicals. However, this practical method is associated with some drawbacks. The chemicals used for the inactivation can affect the structure of surface-exposed antigens, thereby affecting their immunogenic properties. For example, Liu et al. and others observed that chemical inactivation can lead to conformational changes in viral antigens when investigating the effect of the β-propiolactone treatment of a SARS-CoV-2 vaccine candidate [2,3]. In addition, these chemicals might also have some toxic properties, making purification steps necessary during the production process before the vaccine can be applied to humans [4,5]. In this regard, irradiation constitutes a simple and effective method that is broadly used for sterilizing medical devices, food, and household products [6,7]. Furthermore, previously developed vaccines based on irradiated malaria and influenza have confirmed the practicability of this method [8,9]. The same is true for the field of cancer research, where irradiation has been used to generate an anti-cancer vaccine based on inactivated cancer cells, which is being tested in clinical phase II trials (NCT02648282, NCT03006302) [10,11,12]. Nevertheless, γ-irradiation has some drawbacks, including the destruction of surface structures. This in turn has an impact on its ability to induce an adaptive immune response able to confer effective protection. We therefore explore here the potential of accelerated heavy ions as an alternative irradiation method for whole-cell vaccines. We previously showed in a Monte Carlo simulation that γ-rays produce damage in viral surface proteins, whereas the membrane damage caused by a beam of densely ionizing charged particles is significantly reduced. In contrast, the level of RNA lesions caused by both techniques is similar [13]. Thus, a single ion traversal through the nuclear envelope is sufficient to inactivate the virus, at the same time avoiding disruption of the surface structures needed for vaccine efficacy. Following the simulations, we performed an experiment using an influenza virus exposed to ^56^Fe-ions accelerated at 1 GeV/n at the GSI Helmholtz Center SIS18 synchrotron. The irradiated virus was then used to immunize mice by the intranasal route. The obtained results demonstrated that the heavy ion-irradiated intact virions promote both antigen-specific humoral and cellular responses.

## 2. Material and Methods

### 2.1. Mice

C57BL/6 (H-2b) female mice (6–8 weeks old) were purchased from Envigo (Borchen, Germany). All animal experiments in this study were performed with the ethical agreement of the local government of Lower Saxony (Germany) under the permit No. 33.19-42502-04-20/3444.

### 2.2. Virus Inactivation Using Heavy-Ion Beams

The mouse adapted influenza virus A/Puerto Rico/8/1934 were irradiated with heavy ions at the SIS18 synchrotron of the GSI Helmholtz Center in Darmstadt (Darmstadt, Germany). We aimed at using heavy ions with high linear energy transfer (LET) to improve the probability of viral inactivation per traversal, but also high energy to allow a sufficient range in the target to irradiate thick samples containing relatively large volumes through safety containers. We therefore elected to use ^56^Fe-ions 1 GeV/n, whose LET in water is 148 keV/µm and range in water is about 29 cm [14]. To irradiate the samples, we used Eppendorf vials filled with a volume of 100 µL corresponding to approximately 6 × 9 mm^2^ surface (Figure 1a). The Eppendorf tubes were then inserted into Falcon centrifuge tubes and blocked with a plastic ring adapter (Figure 1b). The volume was chosen to ensure a uniform dose to the sample at high intensity, as shown in Figure 1c.

All Falcon tubes, including the virus samples placed on dry ice in a PVC holder, were positioned into a polystyrene box and irradiated in the Cave A of the GSI complex, as shown in Figure 2. All irradiations were performed with spot scanning, using 3 × 4 spots with 4 mm step size (see Figure 1c for the beam spot path in the sample). The samples were exposed at three doses: 5 kGy, 25 kGy and 50 kGy. Since the influenza virus is about 100 nm in diameter, we expected that 0.1% of the virus would be non-hit by any iron ion at 5 kGy, while this fraction would drop below 10^−10^ at 50 kGy. Beam intensity was about 2.5 × 10^7^ ions/s and we needed about 4.2 × 10^9^ ions/cm^2^ to achieve a dose of 1 kGy. With the actual intensity, the irradiation time for 50 kGy was about 5 hrs. Further details on the irradiation methods and dosimetry in Cave A have been published before [15].

### 2.3. Validation of Irradiation Efficacy

Irradiation efficacy was evaluated performing a foci assay. In brief, after propagation of MDCK cells, the cell concentration was set to 6 × 10^5^ cells/mL in the MDCK medium (Minimal Essential Medium (Gibco, Waltham, MA, USA), 10% FCS, 1% penicillin/streptomycin, 1% glutamine) and 100 µL/well was seeded to a 96-microtiter plate. After 24 h of incubation at 37 °C in a 5% CO_2_ atmosphere, serial dilutions of the irradiated sample were added to the MDCK cells. Untreated virus served as the positive control while the cells of the negative control were incubated in the presence of medium only. Afterwards, plates were incubated for 1 h at 37 °C, in a 5% CO_2_ atmosphere. Then, 100 µL of overlay medium (DMEM high glucose, 0.1% MACS BSA solution, N-acetylated trypsin [5 mg/mL], and 1% Avicel) was added to every well and plates were again incubated for 24 h at 37 °C in a 5% CO_2_ atmosphere. After the overlay medium was discarded, and plates were washed twice with PBS before the fixative (4% formalin in PBS) was added. After 10 min of incubation at room temperature (RT), the fixative was discarded and plates washed twice with PBS. In the next step, the quencher (0.5% Triton X-100, glycerin 20 mM in PBS) was added and plates were again incubated for 10 min at RT. Subsequently, the quencher was also discarded; plates were washed once with wash buffer (0.1% Tween 20 in PBS) and blocked (0.1% Tween 20, 1% BSA in PBS) for 30 min at 37° in a 5% CO_2_ atmosphere. Next, the blocking buffer was discarded and plates were incubated for 1 h at RT in the presence of the primary antibody (anti-influenza nucleocapsid goat antibody (1:1000 in blocking buffer). Following three washing steps plates were incubated for 1 h in the presence of the secondary antibody (anti-goat antibody HRP conjugated 1:1000 in blocking buffer). After the final six washes the HRP Substrate (True blue) was added and plates were incubated until blue spots appeared (10 min to 1 h).

### 2.4. Validation of Virus Integrity

Virus integrity following the irradiation process was validated by performing a hemagglutination inhibition assay. Thus, in the first step irradiated virus was incubated overnight at 37 °C with four volumes of receptor destroying enzyme (RDE). Serial two-fold dilutions of RDE-treated virus were incubated for 1 h with 8 hemagglutinating units of PR8 virus before adding 0.7% (*v*/*v*) chicken erythrocytes. After 30 min incubation, the individual hemagglutination inhibition (HI) titers were read as the reciprocal of the highest dilution at which 50% hemagglutination was inhibited. The geometric mean HI titer (GMT) was calculated for each sample and titers < 10 were assigned a value of 5 for calculation purposes.

### 2.5. Immunization Protocols

Female C57BL/6 mice (*n* = 5) were immunized intranasally on day 0, 14 and 28 with vaccine preparations including either chemically inactivated (4 days in the presence of 0.1% formaldehyde) PR8 virus or PR8 irradiated with 50 kGy, both co-administered with the adjuvant bis-(3′,5′)-cyclic dimeric adenosine monophosphate (CDA) in an aqueous solution. The hemagglutinin (HA) protein co-administered with CDA served as the positive control.

### 2.6. Sample Collection

In order to evaluate the stimulated humoral immune responses, blood samples were collected on day −1, 13, 27 and 49 from the retro-orbital complex of the immunized mice. Sera were collected by incubating the blood samples first for 1 h at 37 °C and subsequently for 30 min at 4 °C. Next, samples were centrifuged at 8000× rpm and sera were stored at −20 °C until further processing. On day 49, mice were sacrificed, drained lymph nodes (cervical) and spleens were collected and single-cell suspensions were obtained. To this end, organs were meshed through a cell strainer Falcon 2340 by using a sterile syringe plunger. Cell suspensions were produced and analyzed for the presence of antigen-specific cells, as described previously [16].

### 2.7. Detection of Antigen-Specific Antibodies

PR8-specific antibodies in the sera of individual mice were identified by ELISA using 96-well microtiter plates coated with 2 µg/mL of the antigen (in 0.05 M carbonate buffer, pH 9.6). In brief, coating was performed overnight at 4 °C. After, unspecific binding sites were blocked by incubating the assay for 1 h at 37 °C in the presence of 3% bovine serum albumin (BSA) in PBS. Next, plates were washed several times with 1% BSA/PBS/0.05% Tween 20 (washing buffer) and serial 2-fold dilutions (in 3% BSA/PBS) of sera were added. The secondary antibody (biotinylated goat anti-mouse IgA, IgG, IgG1 and IgG2c (Sigma, Livonia, MI, USA), respectively) was added. Subsequently, samples were again incubated at 37 °C for 1 h. After 1 h of incubation at 37 °C, plates were washed with washing buffer, peroxidase-conjugated streptavidine (BD Pharmingen, San Diego, CA, USA) was added and plates were incubated for 1 h at RT. After another washing step, reactions were developed using the substrate encompassing ABTS [2, 20-azino-bis(3-ethylbenzthiazoline-6-sulfonic acid)] in 0.1 M citrate–phosphate buffer (pH 4.35) containing 0.01% H_2_O_2_. Endpoint titers are expressed as absolute values of the last sample dilution that gave a two-times-higher optical density at 405 nm compared to the blank values. In the case of IgA, results were normalized and expressed as end point titers of antigen-specific IgA per 10 μg/mL of total IgA present in the samples in order to compensate for variations in the efficiency of recovery of secretory antibodies among animals.

### 2.8. Evaluation of Antigen-Specific Cellular Responses

The stimulation of antigen-specific cellular immune responses following the intranasal vaccination of mice was investigated by analyzing the obtained cytokine profiles, as described previously [16]. In brief, supernatants of splenocytes of immunized mice were incubated for 96 h at 37 °C in a 5% CO_2_ atmosphere in the presence of 10 µg/mL of the HA protein. Stimulated cytokine profiles were characterized using the Th1/Th2/Th9/Th17 FlowCytomix immunoassay from Biolegend (CBA), according to the manufacturer’s instructions [16].

In addition, the quantity of antigen-specific cytotoxic CD8+ T cells was investigated using an ELISpot assay, as described previously [17]. In brief, 96-well microtiter plates (BD Pharmingen) were coated with anti-IFN-γ antibodies diluted in PBS and incubated overnight at 4 °C. Blocking of unspecific binding sites was performed incubating the assay for 2 h at RT using complete medium. Following incubation, 4 × 10^5^ and 2 × 10^5^ splenocytes/well of immunized mice were added and incubated in the absence (blank) or presence of the MHC-I immunodominant peptide ASNENMETM [5 µg/mL] of the influenza nucleoprotein (NP). The mitogen concanavalin A [5 µg/mL] served as the positive control. Samples were incubated for 16 h at 37 °C in a 5% CO_2_ atmosphere. After, the assay was washed and incubated in the presence of the biotinylated detection antibody for 2 h at RT. Next, plates were washed prior peroxidase-conjugated streptavidin was added. After one hour of incubation at RT, plates were washed once again and IFN-γ-secreting cells were detected by adding AEC substrate (diluted in 0.1 M acetate buffer pH 5.0 and mixed with 0.05% H_2_O_2_ (30%)). The ELISpot was analyzed using the ImmunoSpot Image Analyzer software v3.2 (CTL-Europe GmbH, Rutesheim, Germany). Results are expressed as Spot Forming Units (SFU) obtained from stimulated cells subtracted by the background from non-stimulated cells [17].

### 2.9. Statistical Analysis

Statistical analysis was performed using the one-way ANOVA test of the Graph Pad Prism 5 software (Version 5.04). Differences were considered significant at *p* < 0.05 (*), *p* < 0.01 (**), and *p* < 0.001 (***), respectively.

## 3. Results

### 3.1. Irradiation Using a Heavy Ion Beam Efficiently Inactivates Influenza Virus

In order to evaluate the energy needed to inactivate the enveloped influenza RNA virus, different energy doses were tested. As shown in Figure 3, only when the virus was exposed to 50 kGy a complete inactivation was obtained. In contrast, energy doses of 5 kGy and 25 kGy did not completely inactivate all viruses in the exposed sample, as indicated by similar infection levels of MDCK cells compared to the untreated PR8 virus (Figure 3).

### 3.2. Irradiation Using a Heavy Ion Beam Maintains Structural Integrity of the Influenza Virus

After demonstrating efficient virus inactivation, the integrity of the overall structure of the surface-exposed viral proteins was evaluated. To this end, a hemagglutination assay was performed. While erythrocytes incubated in the absence of PR8 virus showed the typical sedimentation reaction, building a dot that starts to flow in a thin line once the microtiter plate is lifted, the agglutination reaction observed when erythrocytes were incubated in the presence of irradiated virus disappeared only at a dilution of 1:5120, just as for the untreated PR8 virus (Figure 4). Thus, the obtained results indicate that the structure of the surface protein HA was not affected by the irradiation process, as no differences to the functional properties of the HA of the untreated PR8 virus could be detected.

### 3.3. Immunization of Mice Using Heavy Ion Beam-Inactivated Influenza Virus Stimulated Humoral Immune Responses

In order to investigate the immunogenic potential of the virus inactivated using a heavy ion beam, immunization studies were performed. In this regard, the immunogenicity of our formulation was compared with the classical setting using adjuvanted chemically inactivated influenza virus. At the same time, we wanted to examine whether adjuvantation of virus inactivated using a heavy ion beam was needed in order to obtain sufficient immune responses compared to the classical setting. Therefore, mice were immunized with influenza virus irradiated using a dose of 50 kGy with and without co-administration of CDA. The highest PR8-specific IgG titer was stimulated in mice receiving an equivalent of 10^6^ foci forming units (ffu) of irradiated PR8 co-administered with 10 µg of CDA compared to the blank (*p* < 0.01) (Figure 5a). As demonstrated before, incorporation of the CDA adjuvant resulted in the stimulation of balanced Th1/Th2 responses, as indicated by increased levels of both IgG1 and IgG2c antibodies in sera of immunized mice (Figure 5b) [18]. Inactivated PR8 alone as well as the HA protein co-administered with CDA stimulated a slightly predominant production of IgG1 (IgG1/IgG2c ratio of 3 and 1.9, respectively), indicating a more Th2-biased response. In contrast, the antibody response stimulated by irradiated PR8 virus co-administered with CDA was slightly dominated by the IgG2c isotype (IgG1/IgG2c ratio of 0.5; Figure 5b). Similar results were obtained when analyzing the PR8-specific IgA titers present in lung mucosa following intranasal immunization of mice (Figure 5c). Although not statistically significant, there was a clear trend for a slightly increased IgA titer in mice immunized with irradiated PR8 virus co-administered with CDA, with respect to those observed in mice immunized with adjuvanted HA protein.

### 3.4. Immunization of Mice with Influenza Virus Inactivated Using Heavy Ion Beam-Stimulated Antigen-Specific Cellular Immune Responses

Besides the stimulation of antibody responses, efficient vaccines against viral infection should also stimulate both T helper cells and cytotoxic T cells able to support antibody production and kill infected cells, respectively. In this regard, the data presented here demonstrate that administration of CDA-adjuvanted PR8, inactivated using a heavy ion beam, stimulated a mixed Th1/Th2/Th17 response. Thus, elevated titers of the Th1 cytokines IFN-γ and TNF-α were stimulated following intranasal vaccination as compared to those obtained using formalin-inactivated PR8 co-administered with CDA (Figure 6). In addition, the application of irradiated PR8 co-administered with CDA also stimulated the highest IL-4 titers, whereas the titers of the other investigated Th2 cytokines (IL-5, IL-6 and IL-13) were similar to those observed after restimulation of splenocytes from mice receiving the chemically inactivated PR8 together with CDA (Figure 6). Similar results were obtained when analyzing the levels of the Th17 cytokines IL-17A and IL-22. However, while only marginal differences in the values of IL-17A were obtained comparing chemically and irradiation-inactivated PR8, IL-22 production seems to be slightly increased in mice immunized with irradiated PR8 compared to the formalin-inactivated formulation (Figure 6). Nevertheless, except for IL-4, immunization with the purified HA protein co-administered with CDA stimulated the strongest cytokine production. In contrast, administration of irradiated PR8 alone stimulated only marginal levels of IFN-γ and the production of the Th2 cytokines IL-5 and IL-6. The lower titers produced by splenocytes of mice immunized with whole viral particles might be explained by the cellular response not only directed against a single but many antigens.

Furthermore, immunization of mice by the intranasal route with irradiated PR8 also stimulated antigen-specific CD8+ T cell responses. Significantly increased numbers of IFN-γ-producing T lymphocytes recognizing the CD8 peptide ASNETNMETM of the nucleoprotein in the context of a MHC I molecule were determined in mice receiving irradiated PR8 alone or in combination with CDA with respect to the group immunized with formalin-inactivated PR8 (*p* ≤ 0.001) (Figure 7).

## 4. Discussion

The COVID-19 pandemic clearly demonstrated that vaccines are a valuable tool for efficiently combating rapidly spreading respiratory infectious diseases, reducing deaths and severe disease outcomes. Several approaches were exploited in order to develop effective vaccines against SARS-CoV-2. Besides the innovative RNA vaccines, viral vector- or protein-based vaccines as well as conventional attenuated or inactivated vaccines were also included in the pipeline [19,20,21]. Whole-cell vaccines usually are more reactogenic but encompass different antigenic structures recognized by immune cells and, in case of attenuated viruses, they mimic better the natural infection process. In contrast, immune responses stimulated by subunit and RNA-based vaccines are directed only against one target structure and therefore tend to be short-lived when the pathogen exhibits a large genetic plasticity, leading to immune escape variants. Consequently, these vaccines often require booster immunizations or even variant-adapted formulations to ensure optimal immune protection and long-lasting immunological memory. In this regard, vaccines based on inactivated viruses offer a good compromise. On the one hand, likewise live-attenuated formulations also represent whole cell-based approaches consisting of many immune targets and, on the other hand, they show a similar safety profile as subunit and RNA-based vaccines. However, since they cannot mimic the natural infection process, inactivated vaccines stimulate often weaker and less broad immune responses as compared to attenuated vaccines.

Inactivation of a virus usually is based on chemical and physical methods. The majority of the inactivated vaccines licensed for human use were chemically inactivated using, for example, formalin [22,23]. Physical methods investigated over the decades include heat treatment, UV- and ionizing radiation (usually γ-ray) [24,25,26]. Regardless of the method used for inactivation, the process should leave the immune target structures of the vaccine antigen intact, as this will define vaccine efficacy. Thus, while chemicals, such as formaldehyde or β-propiolactone, are very viable, they tend to damage the surface structures of the antigen, and in general increase the complexity of the purification process to eliminate toxic residuals [27,28,29]. Furthermore, residuals of chemicals can lead to allergic adverse events [30]. Therefore, inactivation of whole-cell vaccines using radiation is still considered a safe and effective method [24,26]. Radiation damages the RNA or DNA of microorganisms but keeps their surface structures mostly intact, essential for vaccine efficacy [31]. Pre-clinical investigation studies have revealed that γ-irradiated vaccine candidates against influenza virus [32], HIV [33], rotavirus [34], and polio [35] stimulate efficient immune responses. Nonetheless, high doses of ionizing radiation (γ-ray) can alter surface structures, resulting in reduced vaccine efficacy [22]. In the case of ionizing radiation, radioprotectors such as a reconstituted Mn-decapeptide (MDP) antioxidant complex have been implemented to limit the damage [35,36]. Other alternatives to γ-irradiation rely on low- [37,38] and high-energy [39] electrons. While low-energy electrons are safer in terms of radioprotection compared to high-energy electrons, which require large and complex shielding to prevent release of radioactive radiation, their impact level is limited, making large-scale vaccine production impossible [24].

In this regard, we here demonstrated that using a beam of high-energy heavy ions constitutes a promising tool to circumvent many of the problems connected to the technologies used so far. Compared to low-energy electrons, high-energy heavy ions have a long impact level of over 25 cm in water-equivalent materials, but at the same time, a reduced attenuation compared to γ-rays and electrons. Moreover, high-energy heavy ions also efficiently inactivate viruses by damaging their nucleic acid but causing only very limited destruction of surface structures targeted by immune cells [40,41,42]. In order to validate our recent results obtained performing a Monte Carlo simulation, we performed in vitro and in vivo experiments demonstrating the potential of this technology [13]. More specifically, influenza A/Puerto Rico/8/1934 was inactivated using a Fe 1 GeV/n beam. Similar to what was shown for other radiation methods, only a high dose of 50 kGy was able to inactivate the virus, as demonstrated by infection experiments using MDCK cells [31,43]. As expected, radiation using a heavy ion beam does not seem to crucially damage immune targets, such as the HA, as proven by a hemagglutination assay. Thus, similar hemagglutination capacities of the irradiated virus and the untreated virus have been observed. Moreover, when immunizing mice with the irradiated virus, strong humoral and cellular immune responses were stimulated. The obtained IgG titers were comparable or even superior to those stimulated using both gamma-irradiated and formalin treated influenza vaccine [44,45,46]. Hence, Chen et al. demonstrated that intranasal vaccination of mice using γ-irradiated influenza virus stimulated a mean antigen-specific IgG titer of about 1:20,000 that is similar to what we have obtained using the heavy ion beam [44]. In contrast, when Singelton et al. immunized mice once with about a 100-fold-increased virus titer they observed a ~20-fold-lower serum IgG titer only [46]. Similar results were obtained by David et al. by immunizing mice two times by the intranasal route using 6-fold increased titers of γ-irradiated virus [47]. In addition, not only surface-located immune targets have been maintained, but also critical epitopes located inside the viral particles. In fact, CD8 T cells were stimulated following vaccination of mice with the irradiated virus, recognizing the MHC class I-restricted ASNETNMETM peptide of the nucleoprotein. Notably, this finding is in line with the finding of Motamedi Sedeh et al. that cell-mediated immune responses stimulated following immunization with irradiated vaccine candidates were prior to those observed following vaccination using chemically inactivated viruses [48]. Similar results were obtained when chickens were immunized to compare irradiated and formalin-treated vaccine candidates [43]. Like the stimulation of efficient antibody responses, this is also of benefit for vaccine-conferred protection. Del Campo et al. demonstrated that protection not only against homosubtypic but also heterosubtypic influenza strains is mediated by nucleoprotein-specific CD8+ T-cells present in the lung and spleen [49]. Furthermore, cellular immunity, especially based on cross-reactive tissue-resident memory T cells, seems to be the key factor for developing a universal influenza vaccine [50].

Taken together, the presented data support the worth of developing whole cell vaccines inactivated using irradiation-based processes. The use of irradiation will also render dispensable cumbersome approaches to remove potential residual toxic moieties [31,51]. Thus, during the SARS-CoV-2 pandemic, several vaccine candidates inactivated using irradiation technologies were tested [26,52,53,54,55]. In addition, also from a commercial point of view, technologies such as electron-based beams (eBeam) or X-ray seem to be attractive tools due to the overall lower costs. Moreover, these technologies can be simply switched off when not in use, whereas radioactive isotopes used for γ-irradiation cannot [24]. The safety profile of inactivated vaccines makes them especially useful for immunocompromised persons. Furthermore, our study demonstrated that the intranasal vaccination of mice using low dosages of irradiated virus co-administered with a mucosal adjuvant stimulates strong immune responses at both a local and systemic level. This might represent an asset in terms of protecting not only against disease, but also against infection, thereby reducing the risk of horizontal transmission to susceptible contacts, a major asset for respiratory pathogens with pandemic potential.

## Figures and Tables

**Figure 1 pharmaceutics-16-00465-f001:**
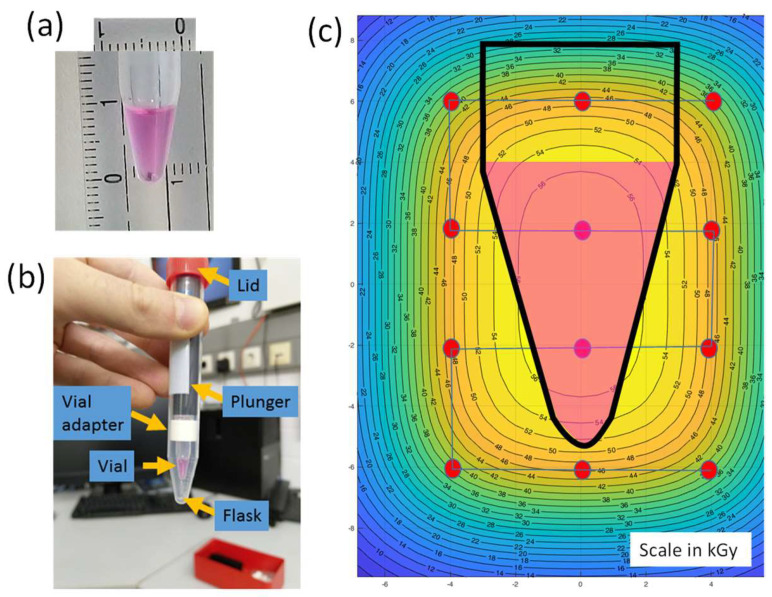
Viral samples for irradiation experiment. (**a**) The Eppendorf tube filled with a volume of 100 µL (scales are in mm). (**b**) To prevent any leaking, the Eppendorf tube is then inserted into a Falcon centrifuge tube. (**c**) Simulated spot scanning path and isodose curves.

**Figure 2 pharmaceutics-16-00465-f002:**
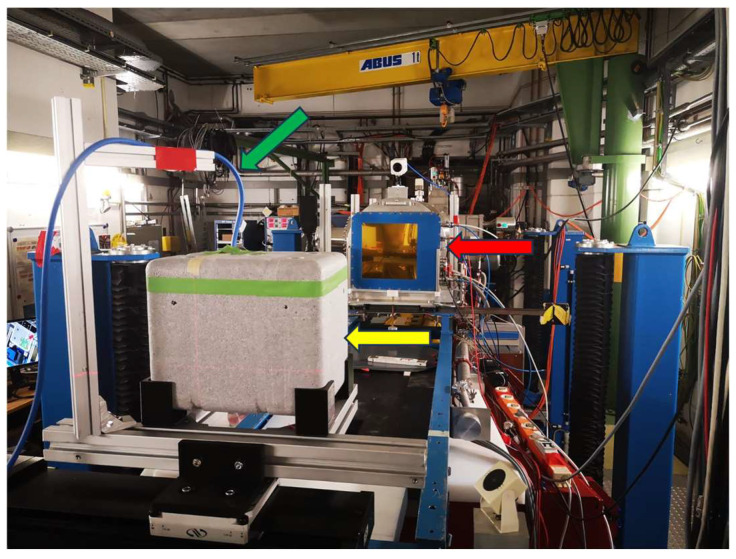
Virus samples in the Styrofoam box irradiated in Cave A. Red arrow points to the monitor chambers. The beam is coming from the monitors toward the target box (yellow arrow). The green arrow shows the cable connected to the pinpoint ionization chamber used to measure the dose at the target position.

**Figure 3 pharmaceutics-16-00465-f003:**
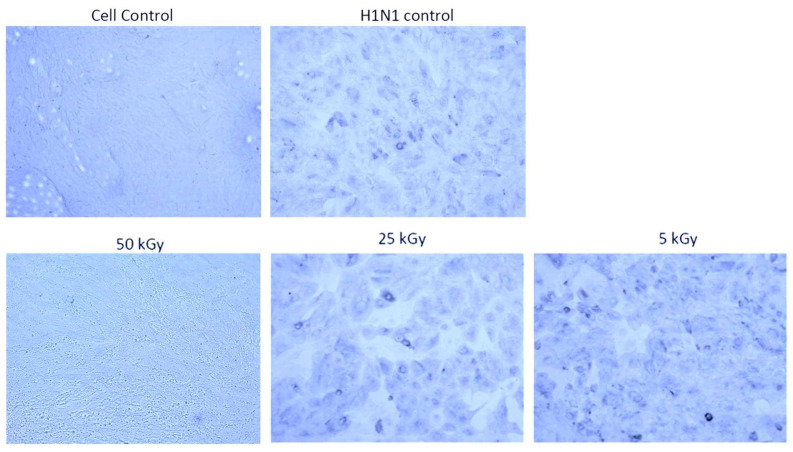
Inactivation of Influenza virus A/Puerto Rico/8/1934 using a Fe 1 GeV/n beam. MDCK-cells were incubated in the presence of virus irradiated with different energy at 37 °C in a 5% CO_2_ atmosphere. Cells of the negative control were incubated in the presence of medium only, whereas untreated virus served as positive control. Only energy of 50 kGy was able to inactivate the virus.

**Figure 4 pharmaceutics-16-00465-f004:**
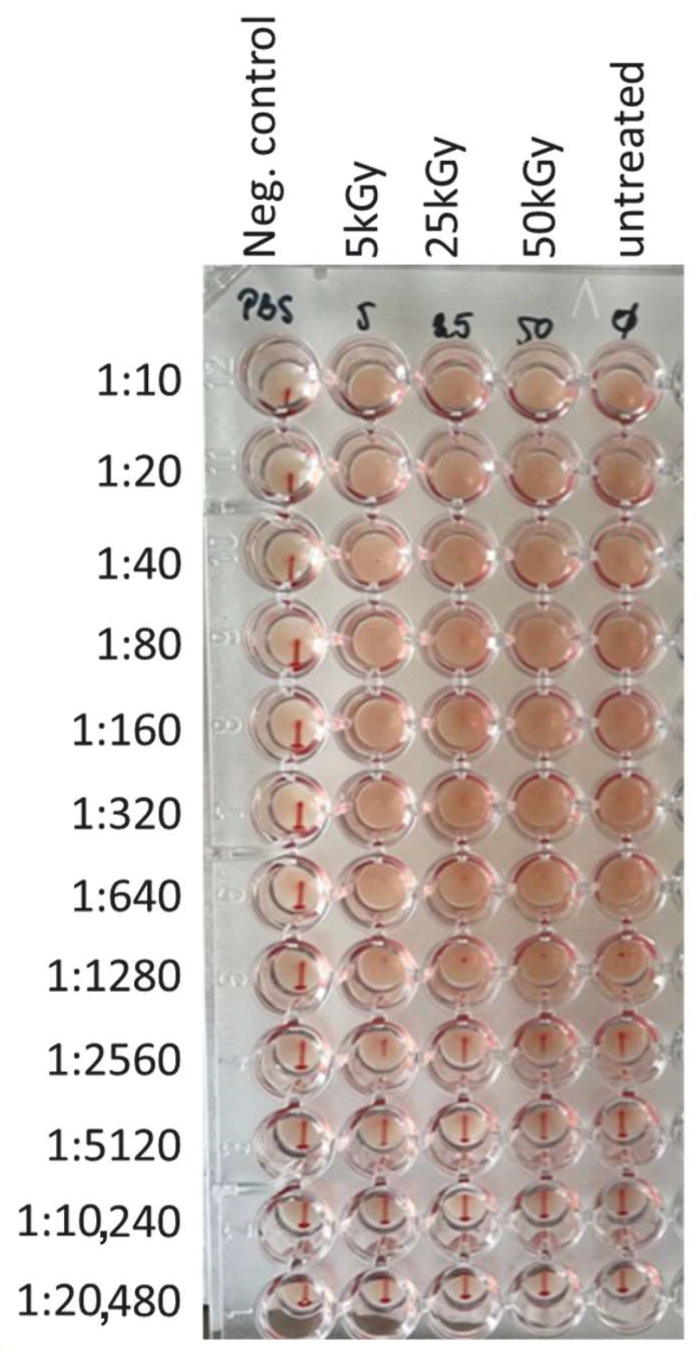
Irradiation of Influenza virus maintains the surface structure. Independently of the used energy irradiation with the Fe 1 GeV/n beam, the hemagglutination capacity of the treated influenza virus was not affected. Only when samples of treated and untreated virus were diluted 1:5120, the viral concentration was too low to allow hemagglutination.

**Figure 5 pharmaceutics-16-00465-f005:**
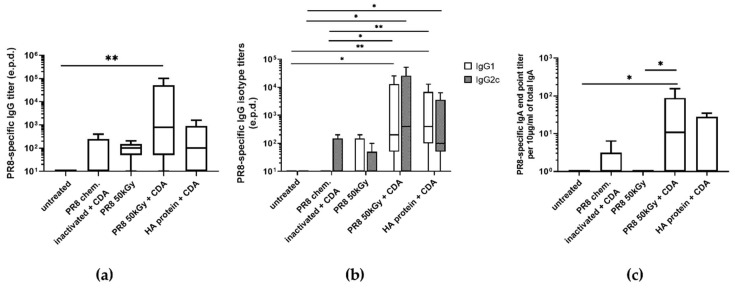
PR8-specific antibody responses stimulated after intranasal immunization of mice with the irradiated Influenza virus A/Puerto Rico/8/1934. PR8-specific IgG titers in sera (**a**,**b**), and secretory IgA titers in lung lavages (**c**) of immunized mice 38 days after the first immunization. IgG titers are expressed as the last dilution (end point dilution, e.p.d.) giving the double value (OD450 nm) of the background value (negative control), whereas IgA titers were normalized and expressed as end-point titers of antigen-specific IgA per 10 μg/mL of total IgA present in the samples. (**b**) PR8-specific IgG subclasses in sera of immunized mice. Results are expressed as the ratio between the IgG1 and IgG2a titers. The standard error of mean (SEM) is indicated by vertical lines. Differences between groups were considered significant at *p* < 0.05 (*) and *p* < 0.01 (**).

**Figure 6 pharmaceutics-16-00465-f006:**
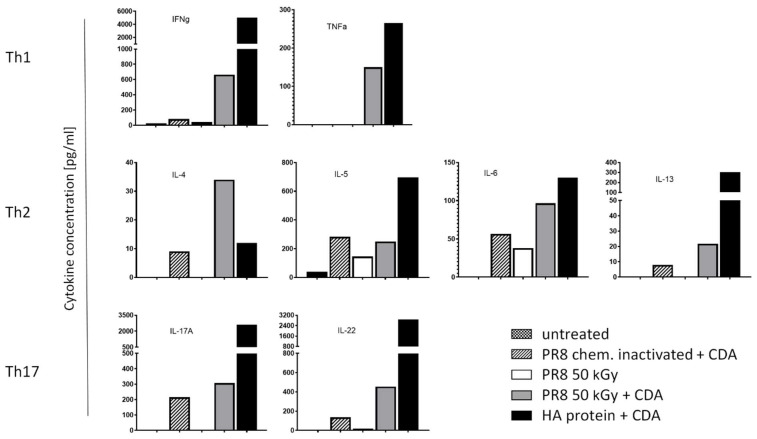
Cytokine profiles stimulated by irradiated Influenza virus A/Puerto Rico/8/1934 co-administered with the CDA adjuvant. The presence of mouse IL-4, IL-5, IL-6, IL-13, IL-17A, IL-22, IFN-γ, and TNF-α were determined in an immunoassay using a cytometric bead array (CBA) according to the manufacturer’s instructions (Mouse Th1/Th2/Th9/Th17 13plex Biolegend). Cytokine concentrations are presented as cluster of Th1, Th2 and Th17 cytokines in HA-restimulated splenocytes derived from mice vaccinated with Influenza virus A/Puerto Rico/8/1934, inactivated by chemical treatment and irradiation, respectively.

**Figure 7 pharmaceutics-16-00465-f007:**
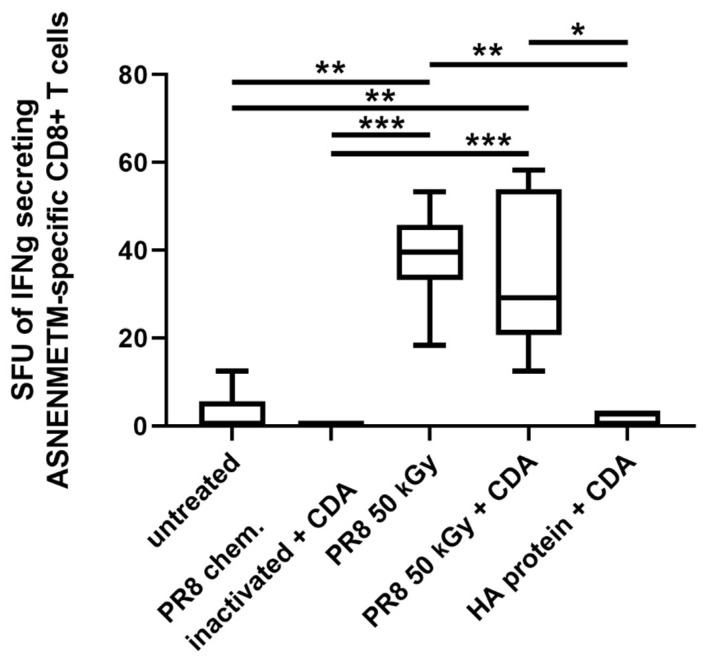
Antigen-specific cytotoxic CD8+ T cells. The number of IFN-γ-producing cytotoxic CD8+ T cells was determined by ELISpot assay. Results are presented as spot-forming units of ASNENMETM-specific (influenza nucleoprotein, NP) cells minus the background values from unstimulated cells. The mean + SD are shown from *n* = 5 animals per group. Standard deviation is indicated by vertical lines. Differences between groups were considered significant at *p* < 0.05 (*), *p* < 0.01 (**) and *p* < 0.001, (***).

## Data Availability

The data presented in this study are available in this article.

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
