# Peer review of "Influenza Virus Inactivated by Heavy Ion Beam Irradiation Stimulates Antigen-Specific Immune Responses"

_pharmaceutics, 2024, doi:10.3390/pharmaceutics16040465_

Round 1

Reviewer 1 Report

Comments and Suggestions for Authors

The authors have studied a way how to improve virus inactivation by using heavy ions. In comparison to most of the vaccines used today, that are chemically inactivated, the heavy ion inactivation comes with the benefit of no need for extra removal of residual chemicals and with better preservation of the antigen target structures at the surface. The latter should be also true when compared to standard radiation method inactivation such as with gamma rays. For the inactivation the authors used 56Fe ions at 1 GeV/n and for the target the mouse adapted influenza virus A/Puerto Rico/8/1934. They showed that the immune targets at the surface were spared as well as the epitopes inside the viral particles. This resulted, after administration to mice, in stimulation of humoral and cellular immune response in the animals. This brings a new knowledge to the field and while these ions are not commonly available, the possible benefits are worthwhile to continue this research. With further research and future improvements this could make a viable method for specific use cases.

Comments

-          Line 117 – “fraction below 10-10” is true for 25 kGy or 50 kGy, please clarify

-          At several places (e.g. Line – 168) and Figures (e.g. Figure 5a) ) – should be “50 kGy” instead of “50 Gy”, as the dose used for virus inactivation? Please check it throughout the manuscript

-          Line 273 – „ffu“ abbreviation not explained

-          Figure 6) – caption: “co-administered with the CDA adjuvant”, but then the legend specifically says CDA at some samples while at others no. The “PR8 chem. Inactivated” was also administered with CDA, correct? While only the “PR8 50 Gy” was without CDA? It should be made clearer.

-          “high energy of 50 kGy” – please change it to: “high dose of 50 kGy”, or similar – it is not energy

Discussion

-          Could a more direct comparison to gamma radiation inactivated vaccines be discussed? (e.g. at line 415 expanded) – not sure how big is the advantage of using the heavy ions; no data for gamma irradiation used / shown (e.g. comparison of the antigen surface structures damage with the use of different types of radiation).

-          The antigen-specific CD8+ Tcell response should be discussed in more detail – it means that “critical epitopes located inside the viral particles” are destroyed by both gamma radiation and chemical inactivation?

Questions

-          The virus was irradiated on dry ice? (could be stressed out more)

-          Iron beam – in the plateau before the Bragg peak – did you consider irradiation in Bragg peak / SOBP to try even higher LET (and/or lower irradiation time)? (possibility of the overkill effect though)

Minor details (typos etc.)

-          Strange sign instead of „gamma“ and „µ“

-          Line 102 – 100 µl

-          Figure 1c) – axes x and y are in mm?

-          Figure 2 – „blue arrow“ is actually green in the image :)

-          Line 186 – parenthesis „)“ missing

Comments on the Quality of English Language

Minor improvements needed.

Mainly just typos, strange signs for um or g-rays, etc...

Author Response

1. Summary

Thank you very much for considering our manuscript for publication. We would also like to thank you for your diligent work and very helpful comments, which we consider instrumental to increase the impact and quality of our manuscript. Please find enclosed a point by point response to each comment below.

2. Point-by-point response to Comments and Suggestions for Authors

Comments 1: Line 117 – “fraction below 10-10” is true for 25 kGy or 50 kGy, please clarify

Response 1: Thank you for pointing this out. We changed the sentence as follows: Since the influenza virus is about 100 nm diameter, we expected that 0.1% of the virus were non-hit by any iron ion at 5 kGy, while this fraction drops below 10-10 at 50 kGy.

Comments 2: At several places (e.g. Line – 168) and Figures (e.g. Figure 5a) ) – should be “50 kGy” instead of “50 Gy”, as the dose used for virus inactivation? Please check it throughout the manuscript

Response 2: Thank you for pointing this out. We corrected the text and figures in the revised manuscript in that the dose for viral inactivation was 50 kGy.

Comments 3: Line 273 – „ffu“ abbreviation not explained

Response 3: Thank you for pointing this out. We explained this abbreviation on page 8, line 276 as follows: The highest PR8-specific IgG titer was stimulated in mice receiving an equivalent of 106 foci forming units (ffu) of irradiated PR8 co-administered with 10 µg of CDA...

Comments 4: Figure 6) – caption: “co-administered with the CDA adjuvant”, but then the legend specifically says CDA at some samples while at others no. The “PR8 chem. Inactivated” was also administered with CDA, correct? While only the “PR8 50 Gy” was without CDA? It should be made clearer.

Response 4: Thank you for pointing this out. We explained more in detail the rational for the chosen immunization groups (page 8, lines 268 ff.): In this regard, immunogenicity of our formulation has been compared with the classical setting using adjuvanted chemically inactivated influenza virus. At the same time, we wanted to examine whether adjuvantation of virus inactivated using a heavy ion beam is needed in order to obtain sufficient immune responses compared to the classical setting. Therefore, mice were immunized with influenza virus irradiated using a dose of 50 kGy with and without co-administration of CDA.

Comments 5: “high energy of 50 kGy” – please change it to: “high dose of 50 kGy”, or similar – it is not energy

Response 5: We corrected the text accordingly (page 12, line 414)

Comments 6: Could a more direct comparison to gamma radiation inactivated vaccines be discussed? (e.g. at line 415 expanded) – not sure how big is the advantage of using the heavy ions; no data for gamma irradiation used / shown (e.g. comparison of the antigen surface structures damage with the use of different types of radiation).

Response 6: We extended the discussion section accordingly comparing more in detail our results with those obtained by others (page 12, lines 422 ff.). The corresponding references were added as well.

Comments 7: The antigen-specific CD8+ Tcell response should be discussed in more detail – it means that “critical epitopes located inside the viral particles” are destroyed by both gamma radiation and chemical inactivation?

Response 7: We extended the discussion section accordingly trying to make clear that chemical inactivation methods seem to impair also immune epitopes located inside the viral particles. Thus, several studies comparing irradiation and chemical methods demonstrated that vaccine formulations based on chemically inactivated pathogens stimulated only weak cell-mediated immune responses with respect to those observed using irradiation techniques (page 13, lines 433 ff.).

Comments 8: The virus was irradiated on dry ice? (could be stressed out more)

Response 8: Yes. In order to clarify this issue we rephrased the sentence as follows: All Falcon tubes including the virus samples placed on dry ice in a PVC holder were positioned into a polystyrene box and irradiated in the Cave A of the GSI complex as shown in Figure 2 (page 3, line 111).

Comments 9:    Iron beam – in the plateau before the Bragg peak – did you consider irradiation in Bragg peak / SOBP to try even higher LET (and/or lower irradiation time)? (possibility of the overkill effect though)

Response 9: This is a very interesting point and we thank the reviewer for this excellent question. The reviewer is certainly right that higher LET would be better. However, Bragg peak would be very difficult from the technical point of view because the penetration depth of the beam would be sub-mm, while it is about 25 cm water-equivalent in the entrance channel. So perhaps the way to go is to use lower energy of iron or even a heavier ion at higher energy - in both cases we wil have higher LET and still a sufficient range to expose thick samples.

Minor details:

-          Strange sign instead of „gamma“ and „µ“

We corrected the symbols accordingly in the revised manuscript.

-          Line 102 – 100 µl

We corrected the symbols accordingly in the revised manuscript.

-          Figure 1c) – axes x and y are in mm?

We add the information in the figure legend.

-          Figure 2 – „blue arrow“ is actually green in the image :)

We corrected the figure legend accordingly.

-          Line 186 – parenthesis „)“ missing

We add the missing parenthesis in line 186.

Reviewer 2 Report

Comments and Suggestions for Authors

Authors demonstrated the high energy heavy ions may be suitable for inactivation of influenza viruses used for vaccination. Heavy ion beams inactivated virus elicited strong antigen-specific humoral and cellular immune response. The manuscript is well written, some misspellings should be corrected. I have minor comments:

-        The spiral sign used for gamma radiation should be replaced by commonly used  γ-radiation.

-        P 108  the symbol   should be replace in the text, for example what is  100l (p108), p100 – 148keV/m  etc. ?

-        P115 what was the dose 5,25  is 5,250 or 5.25?

-        P118   numbers 2.5×107 and  4.2×109 should be write like the other numbers in the text  2.5x107 and 4.2x109

-        P130 how much is 6*105/ml ?

-        Chapter 2.5. Authors should provide information about immunization - the mice were immunized intranasally.

-        P273 what is ffu?

-        Fig6. Figure did not correspond with text. Was chemically inactivated PR8 administrated with CDA or without CDA?

-        Exact values of P should be provided in all pictures.

Author Response

1. Summary

Thank you very much for considering our manuscript for publication. We would also like to thank you for your diligent work and very helpful comments, which we consider instrumental to increase the impact and quality of our manuscript. Please find enclosed a point by point response to each comment below.

2. Point-by-point response to Comments and Suggestions for Authors

Comment 1: The spiral sign used for gamma radiation should be replaced by commonly used  γ-radiation.

Response 1: Thank you for pointing this out. We replaced all spiral signs accordingly throughout the revised manuscript.

Comment 2: P 108  the symbol @ should be replace in the text, for example what is 100 @l (p108), p100 – 148keV/@m  etc. ?

Response 2: Thank you for pointing this out. We replaced all spiral signs accordingly throughout the revised manuscript.

Comment 3: P115 what was the dose 5,25  is 5,250 or 5.25?

Response 3: Thank you for pointing this out. In order to avoid misunderstandings we rephrased the corresponding sentence as follows: The samples were exposed at three doses: 5 kGy, 25 kGy and 50 kGy.  

Comment 4: P118   numbers 2.5×107 and  4.2×109 should be write like the other numbers in the text  2.5x107 and 4.2x109

Response 4: Thank you for pointing this out. We corrected the text accordingly.

Comment 5: P130 how much is 6*105/ml ?

Response 5: We rephrased the text in order to clarify this issue (page 4, line 129): In brief, after propagation of MDCK-cells cell concentration was set to 6x105 cells/ml in MDCK-medium…

Comment 6: Chapter 2.5. Authors should provide information about immunization - the mice were immunized intranasally.

Response 6: Thank you for pointing this out. We add the missing information in the revised manuscript (page 5, line 166)

Comment 7: P273 what is ffu?

Response 7: We explained this abbreviation on page 8, line 276 as follows: The highest PR8-specific IgG titer was stimulated in mice receiving an equivalent of 106 foci forming units (ffu) of irradiated PR8 co-administered with 10 µg of CDA...

Comment 8: Fig6. Figure did not correspond with text. Was chemically inactivated PR8 administrated with CDA or without CDA?

Response 8: We corrected figure 6 adding the CDA adjuvant for group 2 receiving chemically inactivated PR8 virus co-administered with CDA. In addition, we explained more in detail the rational for the chosen immunization groups (page 8, lines 268 ff.): In this regard, immunogenicity of our formulation has been compared with the classical setting using adjuvanted chemically inactivated influenza virus. At the same time, we wanted to examine whether adjuvantation of virus inactivated using a heavy ion beam is needed in order to obtain sufficient immune responses compared to the classical setting. Therefore, mice were immunized with influenza virus irradiated using a dose of 50 kGy with and without co-administration of CDA.

Comment 9: Exact values of P should be provided in all pictures.

Response 9: We decided to not show the exact P value for each relation as this would make the figures confusing. Giving P values of p<0.05 (*), p<0.01 (**) and p<0.001 (***) already underlines the statistically significant differences between the groups. Thus, even if some P values are close to the border of the next level, we do not believe that this influences the core message of our work.

Reviewer 3 Report

Comments and Suggestions for Authors

The manuscript Schulze et al. describes inactivation method of influenza virus using heavy ion beams. The authors proposed that high energy heavy ions may be more suitable as inactivation method, because they increase the damage ratio between the viral nucleic acid and surface proteins. Since inactivation of vaccine using chemicals is associated with some drawbacks, irradiation constitutes a simple and effective alternative method. Present study demonstrated that intranasal vaccination of mice using low dosages of irradiated virus co-administered with a mucosal adjuvant stimulates strong immune responses at both on local and systemic level. So, inactivation of influenza virus with heavy ion beam irradiation is suitable method to develop effective vaccines.

Perhaps, in addition to the advantages of the described inactivation method, the authors should compare the cost of inactivated vaccine by irradiation versus by chemical means. Is it suitable for respiratory infections with pandemic potential from this point of view?

The obtained results are of scientific interest and the manuscript can be published in Pharmaceutics after minor corrections.

Lines 70, 75, 100, 109, 375, 388, 391, 400 invalid symbol.

Line 102: should be mkL

Line 244: “cells” two times.

Author Response

1. Summary

Thank you very much for considering our manuscript for publication. We would also like to thank you for your diligent work and very helpful comments, which we consider instrumental to increase the impact and quality of our manuscript. Please find enclosed a point by point response to each comment below.

2. Questions for General Evaluation

Reviewer’s Evaluation

Response and Revisions

Does the introduction provide sufficient background and include all relevant references?

Yes/Can be improved/Must be improved/Not applicable

Are all the cited references relevant to the research?

Yes/Can be improved/Must be improved/Not applicable

Is the research design appropriate?

Yes/Can be improved/Must be improved/Not applicable

Are the methods adequately described?

Yes/Can be improved/Must be improved/Not applicable

Are the results clearly presented?

Yes/Can be improved/Must be improved/Not applicable

Are the conclusions supported by the results?

Yes/Can be improved/Must be improved/Not applicable

3. Point-by-point response to Comments and Suggestions for Authors

Comments 1: Perhaps, in addition to the advantages of the described inactivation method, the authors should compare the cost of inactivated vaccine by irradiation versus by chemical means. Is it suitable for respiratory infections with pandemic potential from this point of view?

Response 1: Thank you for pointing this out. Therefore, we elaborated more on the cost-benefit relation of both methods (i.e. irradiation versus chemical treatment) in the last paragraph (page 13, lines 444 ff.). In addition, we tried to underline the usefulness of this approach stating that during the corona pandemic several approaches based on irradiation technologies have been investigated in order to develop efficient vaccines against SARS-CoV-2. We add the corresponding references accordingly.

Comments 2: Lines 70, 75, 100, 109, 375, 388, 391, 400 invalid symbol.

 The symbols have been corrected.

Line 102: should be mkL

The unit of measurement has been corrected.

 Line 244: “cells” two times. 

 We deleted one of the two words.
